# Walking with the Atoms in a Chemical Bond: A Perspective Using Quantum Phase Transition

**DOI:** 10.3390/e26030230

**Published:** 2024-03-03

**Authors:** Sabre Kais

**Affiliations:** Department of Chemistry and Elmore Family School of Electrical and Computer Engineering, Purdue Quantum Science and Engineering Institute, Purdue University, West Lafayette, IN 47907, USA; kais@purdue.edu

**Keywords:** quantum phase transitions, finite-size scaling, formation of chemical bond

## Abstract

Phase transitions happen at critical values of the controlling parameters, such as the critical temperature in classical phase transitions, and system critical parameters in the quantum case. However, true criticality happens only at the thermodynamic limit, when the number of particles goes to infinity with constant density. To perform the calculations for the critical parameters, a finite-size scaling approach was developed to extrapolate information from a finite system to the thermodynamic limit. With the advancement in the experimental and theoretical work in the field of ultra-cold systems, particularly trapping and controlling single atomic and molecular systems, one can ask: do finite systems exhibit quantum phase transition? To address this question, finite-size scaling for finite systems was developed to calculate the quantum critical parameters. The recent observation of a quantum phase transition in a single trapped *^171^ Yb^+^* ion indicates the possibility of quantum phase transitions in finite systems. This perspective focuses on examining chemical processes at ultra-cold temperatures, as quantum phase transitions—particularly the formation and dissociation of chemical bonds—are the basic processes for understanding the whole of chemistry.

## 1. Formation of the Chemical Bond

Classical phase transitions, like solid–liquid–gas or order–disorder spin magnetic phases, are all driven by thermal energy fluctuations by varying the temperature. On the other hand, quantum phase transitions happen at absolute zero temperature, with quantum fluctuations causing the ground state energy to show abrupt changes as one varies the system parameters like electron density, pressure, disorder, or external magnetic and electric fields [1]. As a consequence of Heisenberg’s uncertainty principle, the position and the velocity of a quantum object cannot both be measured precisely, causing the ground state energy to show abrupt changes as one varies the system parameters. For example, one can look at the melting of a Wigner lattice (see Figure 1), as suggested theoretically by the physicist Eugene Wigner, a metal that typically conducts electricity could become an insulator when the density of electrons is reduced to ultra-cold temperatures. What has recently been observed experimentally by Park and Demler [2] is the melting of the crystal state into liquid because of quantum fluctuations near absolute zero temperature. These results highlight the beautiful fundamental fact of wave–particle duality in quantum mechanics, wherein electrons behave as “particle-like” in the solid phase, but as “wave-like” in the melted liquid phase. In quantum spin systems, continuous and discontinuous quantum phase transitions have been studied as a function of varying external magnetic fields, pressure, and disorder. In an analogy with the critical point at the classical liquid–gas phase transition in water, Jimenez et al. [3] provided experimental evidence of a quantum critical point in the geometrically frustrated quantum antiferromagnet SrCu_2_(BO_3_)_2_ by controlling both the pressure and the applied magnetic field at low temperatures. These phase transitions are associated with singularities of the free energy that occur only in the thermodynamic limit, where the system size approaches infinity. M.E. Fisher and others developed the finite-size scaling, a systematic theoretical approach that allows one to extrapolate information on the criticality of the finite system to the thermodynamic limit.

However, theoretical and experimental results indicate the existence of quantum phase transitions of finite systems. Plenio and coworkers [4] show that the Quantum Rabi Model (QRM), a two-level spin interacting with a single-mode bosonic field, exhibits a continuous quantum phase transition as one varies the spin-field coupling constant and energies. They prove that in the limit where the spin energy over the field energy tends to infinity, the system has two phases, as follows: the normal phase where the boson field is in the ground state and the superradiant phase where the boson field is excited (see Figure 1). Recently, Cai et al. [5] experimentally observed this continuous quantum phase transition in a system composed of only the two hyperfine atomic levels interacting with the bosonic motional states of an *^171^ Yb^+^* ion in a Paul trap. They measured the spin-up populations in the two hyperfine states in the ground state manifold ^2^S_1/2_ and the rescaled phonon number associated with the spatial motion of the ion along one of its principal axes. The motional degree of freedom can be well described as a quantum harmonic oscillator and thus serve as the Bosonic mode in the quantum Rabi model. This exciting new experimental observation of quantum phase transition in finite systems opens up a new field of research understanding in the quantum critical phenomena of finite systems at ultra-cold temperatures.

Theoretically, the quantum critical phenomena and symmetry breaking of electronic structure configurations of finite systems have been studied extensively for decades. Herschbach and coworkers [6] pioneered the field of dimensional scaling for electronic structure calculations and have discussed the symmetry breaking of electronic structure configurations of both single atoms and simple molecular systems [6]. This approach generalizes the Schrodinger equation to the large-dimensional space and solves the resulting simple equation at the limit D → ∞. At this limit, the electrons take a fixed position in the large-D effective potential, allowing one to study the criticality and symmetry breaking as one varies the parameters controlling the system, such as the nuclear charge, interatomic distance in diatomic molecules, and external electric and magnetic fields. For example, symmetry breaking at the large-D limit of the electronic structure configurations of atoms and simple molecular systems has one-to-one mapping to the criticality of mean-field theory, in analogy to the mean-field criticality of fluid and magnetic systems [7]. To carry out the calculations to the physical space, D = 3, we have shown that one can describe such quantum phase transitions of finite systems in Hilbert space with the analogy between the thermodynamic limit and the size of the Hilbert space [8]. To obtain the exact quantum critical parameters, a finite-size scaling can be formulated in the Hilbert space by replacing the number of particles with the number of basis functions used to obtain the exact ground-state wave function (see Figure 2). This approach can be used to describe the symmetry breaking of electronic structure configurations and the quantum criticality of atomic and molecular systems as one varies parameters in the Hamiltonian [9]. Moreover, another approach to quantum phase transition in describing atom–diatomic phase transitions is the use of the inverse participation ratio and the Rényi entropy as other good markers for the critical point [10,11].

Quantum phase transitions are also used as a standard tool in experiments on ultra-cold atoms, particularly to demonstrate the BEC (Bose–Einstein Condensation) and BCS (Bardeen–Cooper–Schrieffer) cross-over [12]. Here, “Feshbach resonance” is used to change the effective interactions of atoms in a controlled way. At “criticality” the two-atom scattering length diverges which gives rise to many interesting phenomena for the interaction of a cloud of atoms. This occurs when the scattering state of an inter-atomic interaction potential is nearly resonant with the bound state of the potential above it. In this case, an atom pair with the resonant scattering energy have a large probability of forming the bound state. The scattering length diverges in this case if the scattering state energy is exactly resonant with that of the bound state. In an ultra-cold gas, the scattering length can be tuned through the resonance by applying an external magnetic field. Thus, the basic characteristics of Feshbach molecule formation are a divergence of the atom–atom scattering length when approaching the transition from the dissociated side and a divergence of the size of the molecule when approaching the transition from the bound-state side [13,14].

Next, we would like to argue that at ultracold temperatures, using optical tweezers that use a highly focused laser beam to hold and adiabatically move the two separate atoms, allows one to watch the formation of the chemical bond. This might describe the formation/dissociation of chemical bonds as a quantum phase transition between a free atomic state phase and a molecularly bonded state phase. In Figure 2 we describe this possibility wherein we have separate tweezers trapping two individual atoms (Figure 2a) when brought close to one another, leading to the formation of a joint bound state through tunnelling (Figure 2b). To gain insight into the process we envision the event in Figure 2c,d as two sharply peaked delta function wells signifying each strongly trapped atomic site. When the effective distance between the wells is below a certain limit we see a new-bonding (orange) and an anti-bonding (blue) molecular state develop. The bonding state has an enhanced electronic density in the region between the two sites which acts as a cohesive bond for molecule formation. By opening such new ways of examining the formation/dissociation of chemical bonds as quantum phase transitions, one might envision a universal classification of chemical systems. To illustrate this approach, we examine the bond formation of a simple generic homonuclear dimer A_2_ formed from two atomic constituents of A. In the bonded configuration, the total ground state electronic energy of the system A_2_ without correction due to vibrational degrees of freedom can be computed from either a simple Hartree-Fock treatment under the Born–Oppenheimer approximation and subsequently refined with advanced post-Hartree-Fock methods to include static and dynamic correlation. Just like in the QRM model, one might construct a fictitious two-level system in this problem wherein one of the ‘levels’ corresponds to the aforesaid bound state energy of A_2_ at the equilibrium bond length and the other being the ground state energy computed similarly for the free atoms of A. The energy separation between the two forms the excitation energy in the two-level system, just like in the QRM. The interaction of this two-level system with the vibrational stretch mode of A_2_ can now be envisioned to include non-Born–Oppenheimer effects. The said stretch mode with its characteristic frequency under harmonic approximation will form the Bosonic reservoir in the QRM, and the coupling parameter g in the QRM will be replaced by electronic–vibrational coupling strength. After mapping the details of the problem to the QRM model, one can find the critical point and see how it is related to the equilibrium distance (R = R_eq_) in the formation of the molecular state of A_2_ (see Figure 3). So far, we have not discussed the electron and nuclear spin interaction in the case of the formation of A_2_, but one can add the spin degrees of freedom to the analysis and examine the hyperfine splitting changes from free atoms to the formation of molecules.

Adding the spin and the rotational degrees of freedom might give an exotic quantum molecular phase. The analysis of the chemical bond formation and breaking as a quantum phase transition at absolute zero temperature is general for any two-level model system interacting with an environment. This, we believe, would shed unforeseen insight into the very process of forming a chemical bond, which is at the heart of chemistry and molecular physics. Further experimentation is required to investigate the idea, using the optically controlled atomic beams we shall discuss in the next paragraph. One can similarly describe the formation of chemical bonds in any dimeric system, resonance in a benzene molecule, two base states for the molecule of the dye magenta, and nitrogen tunneling in ammonia, to name just a few systems. For example, in the symmetry-breaking transition in the ammonia molecule (NH_3_) as a function of the distance of the atoms, the molecule will probably develop from a planar configuration with all atoms in a plane to the “buckled” configuration where the N atom is either on top of or below the plane defined by the 3 H atoms. This symmetry-breaking transition is driven by a vibrational stretch mode and can be viewed as an approximate quantum critical point.

For many decades, the scientific community has witnessed many Nobel prizes being granted for the basic inventions and discoveries in the area of low-temperature physics, the development of methods to cool and trap atoms with laser light, Bose–Einstein condensation in dilute gases of alkali atoms, and recently for groundbreaking inventions in the field of laser physics and the application of optical tweezers. The field of cold chemistry is an exciting field of research, particularly for quantum information and computing science where quantum mechanical wave-like behavior plays a central role, such as superposition, inference, and entanglement [15,16]. With the advancement of this field, recently Liu et al. [17] presented an experimental study where they combined exactly two atoms in a single, controlled reaction. The experimental apparatus traps two individual laser-cooled atoms, sodium and one cesium, in separate optical tweezers and then merges them into one optical dipole trap, forming one molecule. One must emphasize here that such a bond formation from two individual atoms would need the participation of a third body, the photon, which can act as a scavenger for the lost energy and drive the free atom states to an excited molecular state [18].

In the reverse direction, once a bond is formed, excitation from the lowest vibrational level of the ground electronic potential energy surface (PES) into the continuum (corresponding to loosely bound atoms) in the same ground electronic PES can be initiated through a detuned three-level excitation involving vibrational states of excited electronic PESs. Thus, methods of trapping and controlling single atoms to form molecules might open many exciting avenues for research, particularly quantum phase transitions and the formation of chemical bonds and the possibility of creating exotic molecular quantum phases [19,20,21,22,23,24].

In summary, the very process of chemical bond formation, cornerstone to the foundation of chemistry, can be viewed using the perspective of a quantum phase transition, an idea which thereby unifies the two disciplines and paves the road to further research to consolidate this newly emergent intuition.

## Figures and Tables

**Figure 1 entropy-26-00230-f001:**
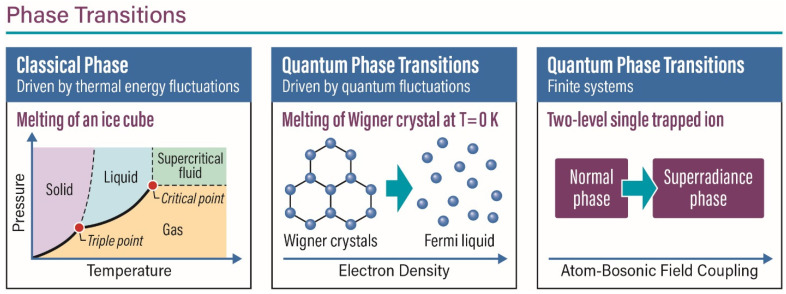
(**Left**) Classical phase transition in a macroscopic system, like water, as characterized in the (T,P) plane. The regimes wherein liquid water, solid ice, and water vapor are stable are represented in three different colors, whereas the black curves indicate phase boundaries wherein two phases can co-exist in equilibrium. (**Middle**) Melting of a Wigner crystal to a Fermi liquid due to increasing electron density. Being a quantum phase transition, such transformations can occur even at absolute zero due to quantum fluctuations. (**Right**) Quantum phase transition (QPT) in a finite system like a single two-level system coupled with an external Bosonic reservoir. Unlike in classical phase transitions wherein macroscopic variables like pressure, temperature, etc., are usually involved, the control variable for a QPT like this is simply the energy separation of the system relative to that of the bath.

**Figure 2 entropy-26-00230-f002:**
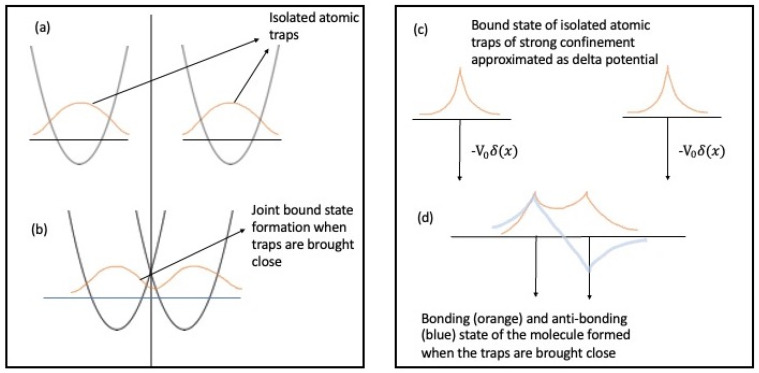
(**a**) Individual atomic sites trapped by a harmonic potential field generated from an optical tweezer. (**b**) Formation of a molecular bound state when the individual traps are adiabatically brought close to one another through the tunnelling of electronic density. (**c**) If the traps are strongly bound one can simplify the process using the model of delta function potential wells. (**d**) Formation of the bonding (orange) and anti-bonding (blue) electronic states when the wells are brought closer to a critical distance.

**Figure 3 entropy-26-00230-f003:**
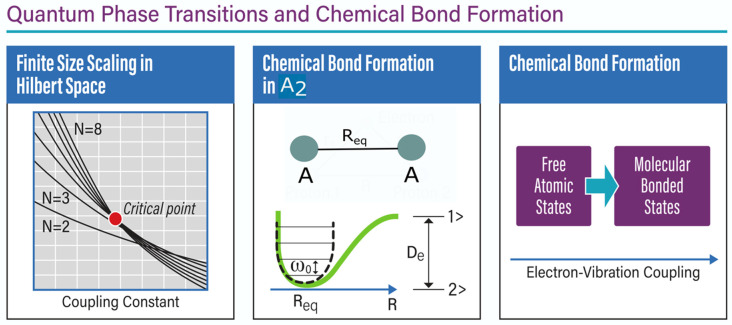
(**Left**) Detection of critical parameters in a quantum phase transition from a finite-size scaling approach. This can be reliably carried out without approaching the limit of infinitely many degrees of freedom, but by successive enhancement of the dimension of the Hilbert space. (**Middle**) The geometric depiction of a generic A_2_ dimer and a single phonon reservoir to which it can be coupled. (**Right**) The two-level description of the chemical bond formation process in A_2_, wherein one of the accompanying states is the free (A,A) units and the other is the bonded system.

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
