# Peer review of "Walking with the Atoms in a Chemical Bond: A Perspective Using Quantum Phase Transition"

_entropy, 2024, doi:10.3390/e26030230_

Round 1

Reviewer 1 Report

Comments and Suggestions for Authors

In the manuscript "Walking with the atoms in a chemical bond: A perspective using quantum phase transition", the author gives an insight into understanding the  formation of a chemical bond from the perspective of a quantum phase transition.

I find this a very interesting idea that is worth exploring. However, I can not
recommend the manuscript for publication in its present form.

There are some issues that should be addressed appropriately:

1) Overall, the paper lacks of a specific structure, i.e. at least an
introduction section, antecedents, idea discussed, and summary.

2) The manuscript is too short, roughly speaking, the main idea is given in one
paragraph: from lines 85 to 114. This idea is then supported experimentally in
the next paragraph: from lines 116 to 133. I think the main idea should be
elaborated much more. Also, there are also papers in the literature describing
the formation of a homonuclear dimer $A_2$ from two atomic constituents of $A$ in terms of a simple two-level system. These previous works should mentioned in this manuscript.

3) The abstract is too long. I think it should be more to the point.

4) from line 38 to 40: "These results highlight the beautiful fundamental
fact of wave-particle duality in quantum mechanics, wherein electrons behave
as “particle-like” in the solid phase, but as “wave-like” in the melted liquid
phase". Please, can the author explain why?

5) The pictures shown in Figures 1 and 2, those related to chemical bond
formation and quantum phase transition, do not give much information as given in the main text. I think the author should include a more illustrative Figure, may be extracted from the paper that the author cites in the manuscript. Also, the rest of pictures could be explained with more detail in the main text of the manuscript.

In summary, I cannot recommend publication of this manuscript unless the above issues are addressed. With some of the improvements given above, I estimate that the article could be reconsidered for publication.

Author Response

I  would like to thank the editors for giving me  an opportunity to respond to the reviews provided by the reviewers. I  also thank the three reviewers for their assessment of our work and for providing critical feedback which was essential for improving the manuscript.

My detailed responses for individual comments from each of the reviewers are described below (Italic)

>>>>>>>>>>>>>>>>> 

  • Comments and Suggestions for Authors: Referee 1

In the manuscript "Walking with the atoms in a chemical bond: A perspective using quantum phase transition", the author gives an insight into understanding the  formation of a chemical bond from the perspective of a quantum phase transition.
I find this a very interesting idea that is worth exploring. However, I can not
recommend the manuscript for publication in its present form.
There are some issues that should be addressed appropriately:

>I thank the referee for his comments to improve this perspective

1) Overall, the paper lacks of a specific structure, i.e. at least an
introduction section, antecedents, idea discussed, and summary.

> In the revised version, I tried to clarify the introduction part and the main idea with the format for a perspective article.

2) The manuscript is too short, roughly speaking, the main idea is given in one paragraph: from lines 85 to 114. This idea is then supported experimentally in the next paragraph: from lines 116 to 133. I think the main idea should be elaborated much more. Also, there are also papers in the literature describing the formation of a homonuclear dimer $A_2$ from two atomic constituents of $A$ in terms of a simple two-level system. These previous works should mentioned in this manuscript.

>Yes, this is a short perspective to illustrate this new connection between the formation of a >chemical bon and quantum phase transition.  Agree with the referee, there are many papers >talking about the formation of different chemical types of  bonds. However since we are >restricted to cite only recent references (last few years!) , I tried to cite a book in the revised >version instead of old references in this field.  

3) The abstract is too long. I think it should be more to the point.

> Agree,  I shortened it in the revised version

4) from line 38 to 40: "These results highlight the beautiful fundamental
fact of wave-particle duality in quantum mechanics, wherein electrons behave
as “particle-like” in the solid phase, but as “wave-like” in the melted liquid
phase". Please, can the author explain why?

>Agree, a good point to be clarify in the revised version which is a results of collaborative wave phenomena as one cooled the system.

5) The pictures shown in Figures 1 and 2, those related to chemical bond
formation and quantum phase transition, do not give much information as given in the main text. I think the author should include a more illustrative Figure, may be extracted from the paper that the author cites in the manuscript. Also, the rest of pictures could be explained with more detail in the main text of the manuscript.

> I tried to explain these figures and illustrate the related physics and keep the short size of this >perspective.

Reviewer 2 Report

Comments and Suggestions for Authors

This is an "opinion" piece on the interpretation of the formation of molecules as a quantum phase transition. This is in principle a valid point of view but, unfortunately, central aspects of this physics are described incorrectly. Furthermore, highly relevant literature in this context is not cited.

Let me start by pointing out that the "quantum phase transition" discussed by the author is used as a standard tool in experiments on ultracold atoms. It is called "Feshbach resonance" and it is used to change the effective interactions of atoms in a controlled way. At "criticality" the 2-atom scattering length diverges which gives rise to many interesting phenomena for the interaction of a cloud of atoms.

The paper says very little about the nature of the transition. It states "The said stretch mode with its characteristic frequency under harmonic approximation will form the bosonic reservoir in QRM and the coupling parameter g in QRM will be replaced by electronic-vibrational coupling strength. After mapping the details of the problem to the QRM model one, the critical point gc=1 might corresponds to the equilibrium distance (R=Req) in the formation of the molecular state of A2 (see Figure 2)."

I do not understand these statements. At the critical point of molecule formation the size of the bound-state wave function (and therefore Req) goes to infinity and the vibronic approximation breaks down completely.

The basic characteristics of molecule formation are a divergence of the atom-atom scattering length when approaching the transition from the dissociated side and a divergence of the size of the molecule when approaching the transition from the bound-state side. None of this is described in this opinion piece. Thus it is not suitable for publication in my opinion.

Further remarks:
- The choice of references on quantum phase transitions is very unusual and selective, I would recommend to cite some of the many review papers on this subject (or the book by Sachdev).

- An early work describing the rather subtle physics of molecule formation in low-density gases from a quantum critical point of view is given in  https://journals.aps.org/pra/abstract/10.1103/PhysRevA.75.033608

Author Response

I  would like to thank the editors for giving me  an opportunity to respond to the reviews provided by the reviewers. I  also thank the three reviewers for their assessment of our work and for providing critical feedback which was essential for improving the manuscript.

My detailed responses for individual comments from each of the reviewers are described below (Italic)

(2) Comments and Suggestions for Authors: Referee2

This is an "opinion" piece on the interpretation of the formation of molecules as a quantum phase transition. This is in principle a valid point of view but, unfortunately, central aspects of this physics are described incorrectly. Furthermore, highly relevant literature in this context is not cited.  Let me start by pointing out that the "quantum phase transition" discussed by the author is used as a standard tool in experiments on ultracold atoms. It is called "Feshbach resonance" and it is used to change the effective interactions of atoms in a controlled way. At "criticality" the 2-atom scattering length diverges which gives rise to many interesting phenomena for the interaction of a cloud of atoms.

> Agree, in the revised version, we clarify the related scattering concept and called "Feshbach resonance". I also added few relevant references as described below

The paper says very little about the nature of the transition. It states "The said stretch mode with its characteristic frequency under harmonic approximation will form the bosonic reservoir in QRM and the coupling parameter g in QRM will be replaced by electronic-vibrational coupling strength. After mapping the details of the problem to the QRM model one, the critical point gc=1 might corresponds to the equilibrium distance (R=Req) in the formation of the molecular state of A2 (see Figure 2)."

I do not understand these statements.

>  I tried to clarify this  description in the revised version  as one think about the transition >between the  free atom phase and the bounded molecular phase. 

The basic characteristics of molecule formation are a divergence of the atom-atom scattering length when approaching the transition from the dissociated side and a divergence of the size of the molecule when approaching the transition from the bound-state side. None of this is described in this opinion piece. Thus it is not suitable for publication in my opinion.

> I agree with the referee, this is very important point to be included and was added to the > >revised version.

Further remarks:
- The choice of references on quantum phase transitions is very unusual and selective, I would recommend to cite some of the many review papers on this subject (or the book by Sachdev).

> This book was added in the original version but  they asked to include only recent reference.  I >agree with the referee as this very important reference and I added it to this revised version.

- An early work describing the rather subtle physics of molecule formation in low-density gases from a quantum critical point of view is given in  https://journals.aps.org/pra/abstract/10.1103/PhysRevA.75.033608

>I agree and was added to this revised version

Reviewer 3 Report

Comments and Suggestions for Authors

This is not a conventional research article. I understand that I have to judge it from the perspective of an opinion piece.

The author argues that some critical chemical processes (like chemical bond formation) can be treated/viewed/explained using the perspective of Quantum Phase Transitions (QPTs) even for a finite (low) finite number of particles. This is a controversial opinion since QPTs traditionally take place in the thermodynamical limit (infinite number N of: particles/components/Hilbert space dimension, etc.).

In fact, classical and quantum phase transitions are concerned with critical/singular behavior for some values of a control parameter (e.g., discontinuities in the derivatives of the energy density with respect to this parameter, according to Ehrenfest's classification) that mark the different phases of the physical system. Criticality usually arises as a consequence of the limit N→infinite. For finite N, abrupt changes are smoothed out and we prefer then to speak of “precursors of the QPT” which announce the emergence of a QPT as the “onset of many-body physics” [see e.g. Nature 587, 583-587 (2020) for recent studies in this direction, although the concept of QPT “precursor” is much older].

There are studies in the literature (like Reference [3] of the manuscript) where authors find critical behavior for a two-level atom in a single mode cavity field (Rabi model) eventhough we are dealing with a single atom (but un unbounded number of photons!). In this case, implicitly there is a limit where the atomic transition frequency, in unit of the cavity frequency, tends to infinity. Again, for finite frequency ratios we should only speak of QPT “precursors”.

To summarize, it makes no sense to talk about water phase transitions (namely ice melting) for ten water molecules. It is true that, in the quantum arena, high quantum fluctuations can lead to quite abrupt changes that denote some degree of "criticality" even for a small number of particles, but I personally would not associate it directly with a QPT. For chemical bond formation processes (the main author’s proposal), I personally prefer the QPT picture of many-body atom-molecule mixtures [like in Phys. Rev. E 90, 042139 (2014)]. I think few-body physics (described by finite-dimensional matrix Hamiltonians) can not describe criticality in the sense of QPTs, although it can provide QPT precursors.

I think the subject is controversial but, judging it as an opinion article, it could be publishable. In any case, I would like to hear an author's response to my report.

Author Response

I  would like to thank the editors for giving me  an opportunity to respond to the reviews provided by the reviewers. I  also thank the three reviewers for their assessment of our work and for providing critical feedback which was essential for improving the manuscript.

My detailed responses for individual comments from each of the reviewers are described below (Italic)

(3) Comments and Suggestions for Authors: Referee 3

his is not a conventional research article. I understand that I have to judge it from the perspective of an opinion piece. The author argues that some critical chemical processes (like chemical bond formation) can be treated/viewed/explained using the perspective of Quantum Phase Transitions (QPTs) even for a finite (low) finite number of particles. This is a controversial opinion since QPTs traditionally take place in the thermodynamical limit (infinite number N of: particles/components/Hilbert space dimension, etc.).

>The referee raised a good point.  However, in our work on quantum phase transition of fine >system, we draw the analogy between the number of basis set used in Hilbert space to the >thermodynamic limit. The  analogy leads to derive finite size  scaling allowing one to calculate >all the critical parameters by taking the limit of infinite size of Hilbert space! This point was >clarify in the revised version .

In fact, classical and quantum phase transitions are concerned with critical/singular behavior for some values of a control parameter (e.g., discontinuities in the derivatives of the energy density with respect to this parameter, according to Ehrenfest's classification) that mark the different phases of the physical system. Criticality usually arises as a consequence of the limit N→infinite. For finite N, abrupt changes are smoothed out and we prefer then to speak of “precursors of the QPT” which announce the emergence of a QPT as the “onset of many-body physics” [see e.g. Nature 587, 583-587 (2020) for recent studies in this direction, although the concept of QPT “precursor” is much older].

>Agree with the referee and thank you for pointing us to this important paper to be included in >the revised version

There are studies in the literature (like Reference [3] of the manuscript) where authors find critical behavior for a two-level atom in a single mode cavity field (Rabi model) eventhough we are dealing with a single atom (but un unbounded number of photons!). In this case, implicitly there is a limit where the atomic transition frequency, in unit of the cavity frequency, tends to infinity. Again, for finite frequency ratios we should only speak of QPT “precursors”.

>Agree and clarify this point in the revised version.

To summarize, it makes no sense to talk about water phase transitions (namely ice melting) for ten water molecules. It is true that, in the quantum arena, high quantum fluctuations can lead to quite abrupt changes that denote some degree of "criticality" even for a small number of particles, but I personally would not associate it directly with a QPT. For chemical bond formation processes (the main author’s proposal), I personally prefer the QPT picture of many-body atom-molecule mixtures [like in Phys. Rev. E 90, 042139 (2014)]. I think few-body physics (described by finite-dimensional matrix Hamiltonians) can not describe criticality in the sense of QPTs, although it can provide QPT precursors. I think the subject is controversial but, judging it as an opinion article, it could be publishable. In any case, I would like to hear an author's response to my report.

>Again, I would like to thank the referee for the critical relevant points and references to add >and to clarify important concepts which I did in the revised version.

Round 2

Reviewer 1 Report

Comments and Suggestions for Authors

Instead of a major revision, the author has carried out a minor revision of the manuscript. I understand that as it is an opinion piece, it will have a word limit. However, the author has not revised the figures, as suggested.

On the other hand, I believe the author has not looked for adequate references, arguing that only includes recent ones. However, in the manuscript there are references from the 1990s. Perhaps the author should reconsider including more recent references for instance:

* Graefe, E.M.; Graney, M.; Rush, A. Semiclassical quantization for a bosonic
atom-molecule conversion system. Phys. Rev. A 2015, 92, 01212.

* Baena, I.; Pérez-Fernández, P.; Rodríguez-Gallardo, M.; Arias, J.M.;
Entropies and IPR as Markers for a Structural Phase Transition in a Two-Level Model for Atom-Diatomic Molecule Coexistence. Entropy 24, 2022.

Overall, I am not completely satisfied with the author's review. I cannot
recommend the article for publication.

Author Response

I have done a major revison 

(1 ) Included the two relevant references. 
* Graefe, E.M.; Graney, M.; Rush, A. Semiclassical quantization for a bosonic
atom-molecule conversion system. Phys. Rev. A 2015, 92, 01212.

* Baena, I.; Pérez-Fernández, P.; Rodríguez-Gallardo, M.; Arias, J.M.;
Entropies and IPR as Markers for a Structural Phase Transition in a Two-Level Model for Atom-Diatomic Molecule Coexistence. Entropy 24, 2022.

(3) Added a few lines 

Moreover, another approach to quantum phase transition in describing  atom-diatomic phase transition is the use of the inverse participation ratio  and the Rényi entropy as other good markers for the critical point[10,11].

(3) Added a new figure (2) to describe the processs

Reviewer 2 Report

Comments and Suggestions for Authors

In response to my previous comments, the author added references and a discussion of the Feshbach resonance.
The physics discussed in the context of the Feshbach resonance shows that one can indeed view molecule formation as a quantum phase transition: upon changing a parameter, one obtains a "phase transition" between dissociated atoms and a molecule. This transition is characterized by a diverging length scale.

The author has, however, (or seems to have) something completely different in mind. In the central section of the paper, he wants to consider what changes when the "electronic-vibrational coupling strength" is changed. Furthermore, he says "the critical point gc=1 might corresponds to the equilibrium distance (R=Req) in the formation of the molecular state".  I have no clue why this should be relevant concept. Is he claiming that each molecule in equilibrium is at a critical point? If one takes a generic molecule and changes the "electronic-vibrational coupling strength" g a bit, nothing qualitatively new happens, the equilibrium distance will just adjust a little bit. Therefore, I simply do not understand what physics the author wants to discuss.

He continues to talk about bond-formation and breaking. Does he have in mind that as function of g one obtains a transition from a bound molecule to seperate atoms? Shouldn't in this case the size of the bound state diverge at the critical point (instead of taking the "equlibrium value" as claimed by the author)?

In conclusion, I still do not understand the central part of the paper. For an "opinon" piece speculations are OK but at least I cannot profit from this paper because I do not understand its central element.

Author Response

modified the text  to clarify  the part which is addition to formation of Feschbach molecules. 

"Using Optical tweezers that use a highly focused laser beam to hold and move adiabatically the two separate atoms allowing one to watch the formation of the chemical bond. This might describe the formation/dissociation of chemical bonds as a quantum phase transition between a free atomic state phase and a molecularly bonded state phase as describe in the new Figure (2) ."

  This process is different from the formation of bond Fescbcah molecule !  this is focused not on collision  but using Optical tweezers and move the two trap atoms adiabatically and then bring them down with a laser to the bounded ground state.  With the hope that one can find an observable to watch the formation of the chemical bond and if one can describe it as QFT !  

Round 3

Reviewer 1 Report

Comments and Suggestions for Authors

I think the paper is suitable to be published.

Author Response

Thank you for your valuable comments to improve the paper. 

Reviewer 2 Report

Comments and Suggestions for Authors

The author has added a new idea: atoms are hold in a tweezer and there is a phase transition as function of their separation. The phase transition is thereby thought to be of the type where a single two-level system couples to a reservoir. Such transitions can be viewed as quantum critical points in certain limiting cases. I now understand a bit better what the author has in mind.

A major problem with the paper is that for the situation described (atoms in tweezers approaching each other) there is typically no (approximate) phase transition as far as I can tell. More precisely, the statement "When the effective distance between the wells is below a certain limit we see a new-bonding (orange) and an anti-bonding (blue) molecular state develop" appears to be wrong (at least at zero temperature): there is no critical distance where the bonding and anti-bonding states develop. For large distances, the bonding-antibonding splitting is just exponentially small but always finite (even when vibrational modes are considered).

There is, however, a variant of the protocol discussed by the author which would work in the spirit of the paper: Take, e.g., an ammonia molecule NH3.
Here as a function of distance of atoms, the molecule will probably develop  from a planar configuration with all atoms in a plane to the "buckled" configuration where the N-atom is either on top or below the plane defined by the 3 H-atoms. This symmetry-breaking transition is driven by vibrational stretch mode and can be viewed as an approximate quantum critical point (because the tunneling rate of N beweent the two configuration is very small) as envisioned by the author.

My suggestion is that the author adds something like that and then have the paper published.

Author Response

Thank you for your valuable comments to improve the paper. 

I added the example of Ammonia as suggested by the referee.